# Nanoscale Superconducting States in the Fe-Based Filamentary Superconductor of Pr-Doped CaFe_2_As_2_

**DOI:** 10.3390/nano11041019

**Published:** 2021-04-16

**Authors:** Giang D. Nguyen, Mingming Fu, Qiang Zou, Liurukara D. Sanjeewa, An-Ping Li, Athena S. Sefat, Zheng Gai

**Affiliations:** 1Center for Nanophase Materials Sciences, Oak Ridge National Laboratory, Oak Ridge, TN 37831, USA; giangnguyen7386@gmail.com (G.D.N.); mmfu@ncu.edu.cn (M.F.); qzou.iphy@gmail.com (Q.Z.); apli@ornl.gov (A.-P.L.); 2Department of Physics, Nanchang University, Nanchang 330031, China; 3Materials Science & Technology Division Oak Ridge National Laboratory, Oak Ridge, TN 37831, USA; lsanjeew@utk.edu (L.D.S.); athena.sefat@science.doe.gov (A.S.S.)

**Keywords:** iron-based superconductor, filamentary superconductor, nanoscale superconducting states, defects, scanning probe microscopy, defects, domain boundary

## Abstract

The low-temperature scanning tunneling microscope and spectroscopy (STM/STS) are used to visualize superconducting states in the cleaved single crystal of 9% praseodymium-doped CaFe_2_As_2_ (Pr-Ca122) with *T_c_* ≈ 30 K. The spectroscopy shows strong spatial variations in the density of states (DOS), and the superconducting map constructed from spectroscopy discloses a localized superconducting phase, as small as a single unit cell. The comparison of the spectra taken at 4.2 K and 22 K (below vs. close to the bulk superconducting transition temperature) from the exact same area confirms the superconducting behavior. Nanoscale superconducting states have been found near Pr dopants, which can be identified using *dI/dV* conductance maps at +300 mV. There is no correlation of the local superconductivity to the surface reconstruction domain and surface defects, which reflects its intrinsic bulk behavior. We, therefore, suggest that the local strain of Pr dopants is competing with defects induced local magnetic moments; this competition is responsible for the local superconducting states observed in this Fe-based filamentary superconductor.

## 1. Introduction

Discovery of the high-*T_c_* of Fe-based superconductors has attracted much attention recently. Among the materials, 122-type iron-based superconductivity is of particular interest due to its relatively simple structure and the easy growth of large single crystals [1]. The parent compound of the 122 family exhibits antiferromagnetic order at low temperature, and the superconductivity typically emerges through chemically doping. However, the mechanism of doping-induced superconductivity in 122 superconductors is still controversial. Additionally, the superconducting doped samples are electronically inhomogeneous, even on a nanoscale. Therefore, a scanning probe microscope has been an ideal tool to study the doping effects as well as the superconducting mechanism on these Fe-based superconductors [2]. The majority of recent research in this area focuses on bulk superconducting samples, and only limited studies have been conducted on filamentary superconducting behavior. It has been argued that the filamentary superconductors arise due to spontaneous electronic inhomogeneity at the nanoscale level of the sample, and STM/S might be the most adequate tool to reveal the insight to filamentary behavior [3,4]. Such electronic inhomogeneities may be caused by nanoscale dopants and disorder, which manifest in non-zero resistance and have a very small Meissner effect (field-cooled in magnetic susceptibility), although their shielding is the same as bulk superconductors [5,6]. As an example of the neglected filamentary superconductor, electron-doped, Pr-doped CaFe_2_As_2_ (Pr-Ca122) crystal shows the highest transition temperature among the pnictides with ThCr_2_Si_2_ crystal structure (*T_c_*~45K). The difficulty in using STM to achieve this goal is how to identify the dopant position and correlate it to superconducting states. Recently, Gogryky et al. studied the optimal-doped Pr-Ca122 crystal (14% dopant) and found the local electronic inhomogeneous; [3] Zeljkovic et al., demonstrated a method for identifying the Pr-dopant location on a filamentary superconductor of Pr-doped CaFe_2_As_2_ (Pr-Ca122) [4]. However, there was no STM spectroscopy performed to correlate this dopant location with superconducting states.

Here, we use scanning tunneling microscopy and spectroscopy (STM/STS) to study local superconducting states on lower Pr-doped Ca122 crystal (9% dopant) to avoid segregated dopant clusters and to corelate filamentary superconductivity to local structure and dopants on a nanoscale.

## 2. Materials and Methods

Single crystals of Pr-doped CaFe_2_As_2_ (Pr-Ca122 were grown out of FeAs self-flux technique, similar to ref. [1,7], with [001] direction perpendicular to the crystalline plate shapes. The chemical composition of the crystals was measured with a Hitachi S3400 scanning electron microscope energy-dispersive X-ray spectroscopy (EDS). The structures were identified as tetragonal ThCr2Si2 type (I4/mmm, Z = 2) at room temperature, and lattice parameters upon doping were refined using X’Pert HighScore by collecting data on an X’Pert PRO MPD X-ray powder diffractometer. Magnetic data were collected using Quantum design’s magnetic property measurement system (MPMS).

Single crystals of Pr-Ca122 were cleaved at liquid nitrogen temperature in an ultra-high vacuum system, then immediately transferred into an in-situ STM precooled to 4.2 K without breaking vacuum. The STM/S experiments were carried out using a scanning tunneling microscope with base pressure better than 2 × 10^−10^ Torr, with a chemically etched W tip. All W tips were conditioned on clean Au (1 1 1) and checked using the topography, surface state, and work function before each measurement. The STM/S were controlled by the SPECS Nanonis control system. Topographic images were acquired in constant current mode with bias voltage applied to samples and tip grounded. All the spectroscopies were obtained using the lock-in technique with a modulation of 1 mV at 973 Hz on bias voltage, *dI/dV*. Current-imaging-tunneling-spectroscopy was collected over a grid of pixels at bias ranges around Fermi level using the same lock-in amplifier parameters.

## 3. Results and Discussion

The crystal structure of the parent compound of Pr-doped CaFe_2_As_2_ (Pr-Ca122) is shown in Figure 1a. It composed of a trilayer building block of FeAs sandwiched between checkerboard layers of Ca, with lattice constants of a = b = 0.395 nm and c = 1.3 nm. A small amount (9% measured concentration in this work) of Pr is used to dope the compound by substituting at the Ca site. The temperature dependence of magnetic susceptibility in the zero-field cool measurement of the Pr-Ca122 sample is shown in Figure 1b. The downturn of the susceptibility signifying diamagnetic responses as a result of the Meissner state reveals a superconducting transition at 30 K. However, the sample is only considered a filamentary superconductor due to its small superconducting volume fraction (~3% at 5 K) [3,8,9,10]. In order to visualize the local superconductivity behavior in Pr-Ca122, we performed a STM study on in situ cleaved samples. Figure 1c shows a typical large-scale STM morphology image of the cleaved surface. The step height of ~0.65 nm (line profile in the inset) is about a half of the unit cell size along the c axis. It reflects the mirror symmetry of Ca122 crystal structure (Figure 1a).

An atomic resolution STM image (Figure 2a) demonstrates a stripy pattern with a periodicity of 0.80 ± 0.02 nm, which is the double unit cell in the ab plane. It suggests the surface is terminated by Ca atoms with a 2 × 1 reconstruction [4]. To identify the Pr dopants on the surface terminated by Ca atoms, we applied a similar method discussed in the previous report [4]. The Pr dopants appear as a bright area in a *dI/dV* conductance at a high positive sample bias of 300 mV (marked by dashed white rings in Figure 2b). The *dI/dV* maps at the negative sample biases (Figure 2c,d) do not show any distinctive contrast.

As the global susceptibility result of Pr-Ca122 shows filamentary superconducting behavior, it is interesting to reveal how filamentary superconductivity corelates with local structures. When surveying around the samples, we find that although STS *dI/dV* curves show gap-like features in occasional spots, STS from majority areas do not show superconducting gap. To visualize the local superconductivity, a superconducting gap map serves better than an STS *dI/dV* map. As an example, Figure 3 shows a STM topographic image and corresponding superconducting gap map, which is calculated from the current imaging tunneling spectroscopy (CITS) image at 4.2 K [11,12]. The majority area of the sample does not show superconductivity, which is consistent with a low superconducting volume fraction from the bulk measurement (about 3%). The localized superconducting areas are observed mostly near Pr dopants, which we can identify as Figure 2b (Figure 3b, white rings). Surprisingly, the superconducting state is very localized, which can be in the range of a few cell-size units of the compound (a = 0.395nm). The typical extracted *dI/dV* spectra across a superconducting region (blue arrow in Figure 3a) are presented in Figure 3c. The spatial distance between each curve is 0.12 nm. The five red curves show superconducting states, which have a superconducting symmetric gap at Fermi level with coherent peaks. Away from the superconducting region, the *dI/dV* spectra show normal metallic states (black curves). In between the red and black curves, there are a couple of curves that show pseudogap states (green curves); they are the mixing between superconducting and normal states [12]. The local superconducting area at this position is around 1 nm.

There are some areas on the surface with higher Pr dopant concentrations, mostly around defective regions and domain boundaries. Figure 4a,b show the STM topographic image and *dI/dV* map, respectively, in such a region, with Pr dopants (yellow spots and lines) along bulk domain boundaries and nearby strained areas. Figure 4c displays the *dI/dV* line spectra at a temperature of 6 K along a black arrow in Figure 4a. The curves show a gap-like feature, which presents the superconducting and pseudogap phase. At an increased temperature of 22 K, the *dI/dV* line spectra recorded in the sample region show strongly suppressed superconducting gap states (Figure 4c). This is consistent with the bulk measurement of the superconducting transition temperature of the sample at about 30 K (Figure 1b).

It is important to note that the domain boundaries and defective regions where Pr dopants segregate are intrinsic bulk-like structures, which are created during crystal growth. Those sites are different from the 2 × 1 surface reconstruction 90-degree domain boundaries, as pointed out by a red arrow in Figure 4a. As shown in Figure 4b, Pr dopant concentration near reconstruction domain boundaries is similar to regular areas, as these boundaries are the result of restructuring surface Ca/As/Pr atoms during exposure of the surface at cleavage.

We also notice the local superconductivity observed in this work is not originated from surface-related defects. Shown in Figure 5a are some examples of local defects. The STS map of the area shows no Pr dopants around. The STS *dI/dV* spectra for defects 1, 2, 3 and the non-defected area are shown in Figure 5b. Although defects areas are different from the pristine area, none of them show superconducting gap or pseudogap features.

Here we discuss the role of Pr dopants in response to local superconductivity in Pr-Ca122. This observation of local superconducting states near Pr dopants supports the hypothesis of the superconducting phase induced by local strains near Pr dopants [13,14,15,16]. The local strain can yield the optimum Fe-As bonding angle for emergence of the superconducting phase [17]. However, these Pr dopants, in parallel, create lattice defects with some net magnetic moment [18]. The local magnetic moment can also destroy superconductivity [11]. Therefore, the Pr-Ca122 sample is not able to become a bulk superconductor by simply increasing the Pr dopant concentration. These two competing processes result in the filamentary superconductivity observed in Pr-Ca122.

## 4. Conclusions

Low-temperature STM/STS is used to observe nanoscale superconducting states in the cleaved single crystal of low Pr-doped CaFe_2_As_2_. The spectroscopy shows strong spatial variations of density of states, and a superconducting map constructed from spectroscopy discloses a localized superconducting phase, as small as a single unit cell. The comparison of the spectra taken at low temperature (4.2 K) and an elevated temperature 22 K, which is below but near the bulk transition temperature area, confirms superconducting behavior. Nanoscale superconducting states have been found near Pr dopants, which can be identified using *dI/dV* conductance maps at +300 mV. Local superconductivity is not influenced by surface reconstruction and surface defects created during cleavage. We, therefore, suggest that the competition between the local strain at Pr dopants and the local magnetic moment induced by defects is responsible for the local superconducting states observed in this Fe-based filamentary superconductor.

## Figures and Tables

**Figure 1 nanomaterials-11-01019-f001:**
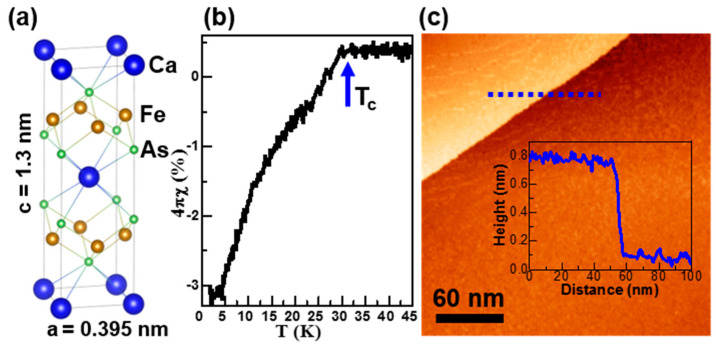
(**a**) Crystal structure of Ca122. (**b**) Temperature dependence of zero-field-cooled magnetic susceptibility of the sample of Pr-Ca122, showing diamagnetism below ~30 K, under a constant magnetic field of 20 Oe. (**c**) Large scale STM topographic image after cleaving (sample bias V_S_ = −20 mV, tunneling current *It* = 100 pA, temperature *T* = 4.2 K). The line profile across a half unit cell step is shown at the inset.

**Figure 2 nanomaterials-11-01019-f002:**
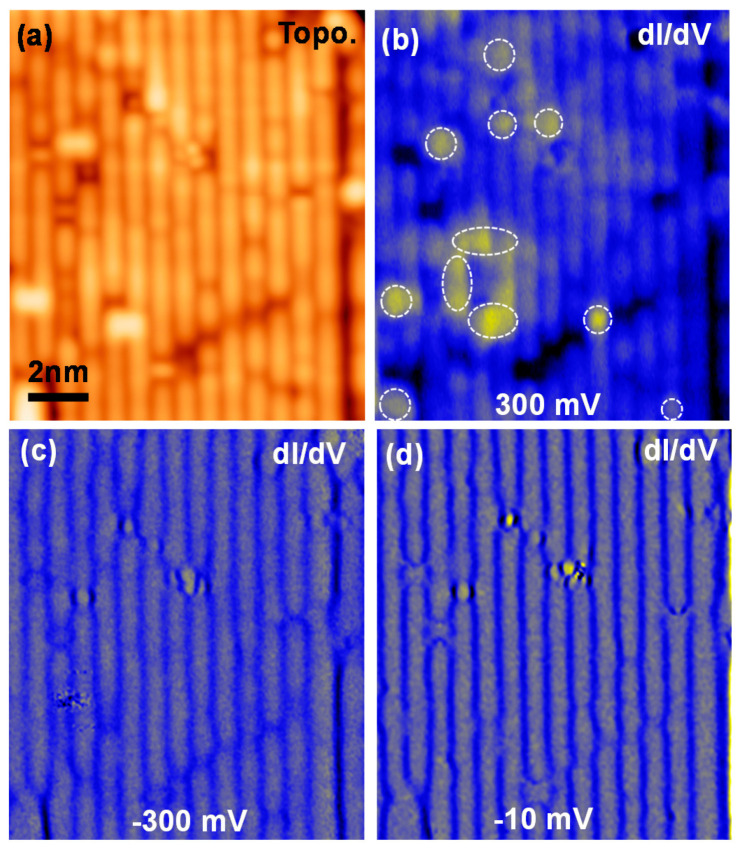
(**a**,**b**) STM image and *dI/dV* conductance map on Pr-Ca122 (Vs = 300 mV, It = 200 pA, modulation voltage *Vac* = 1 mV, *f* = 973 Hz, *T* = 4.2 K). The bright spots on the *dI/dV* map in (**b**) indicate the Pr-dopant positions (dashed white circles). (**c**,**d**) Same area with *Vs* = −300 mV and −10 mV, respectively.

**Figure 3 nanomaterials-11-01019-f003:**
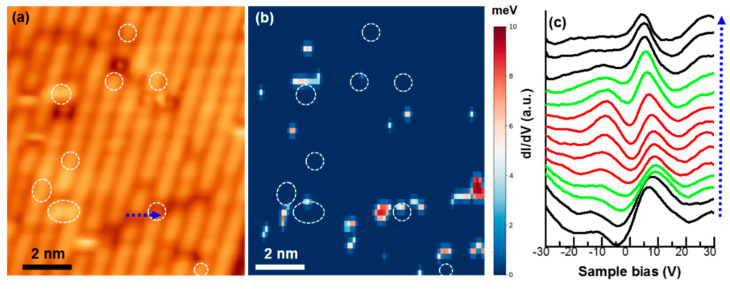
(**a**,**b**) STM image and superconducting gap map respectively extracted from CITS measurement at 4.2 K on the surface area, as shown Figure 2a (*V_S_* = −10 mV, *It* = 200 pA, *V_ac_* = 0.5 mV, *f* = 973 Hz, *T* = 4.2 K). The white dashed rings mark the position of Pr dopants, as found on the *dI/dV* map in Figure 2b. The distortion of the image compared with Figure 2 is due to the thermal drift of 15 h CITS. (**c**) *dI/dV* line spectra taken across a superconducting phase area marked by the blue arrow in (**a**). The spatial distance between each taken curve is 0.12 nm. Red and black mark the superconducting density of states and normal metallic density of states, respectively. The transition curves from the superconducting density of states to the normal metallic density of states have pseudogap characteristics, which are marked with a green color. Vertical offsets are applied to the spectra for clarity.

**Figure 4 nanomaterials-11-01019-f004:**
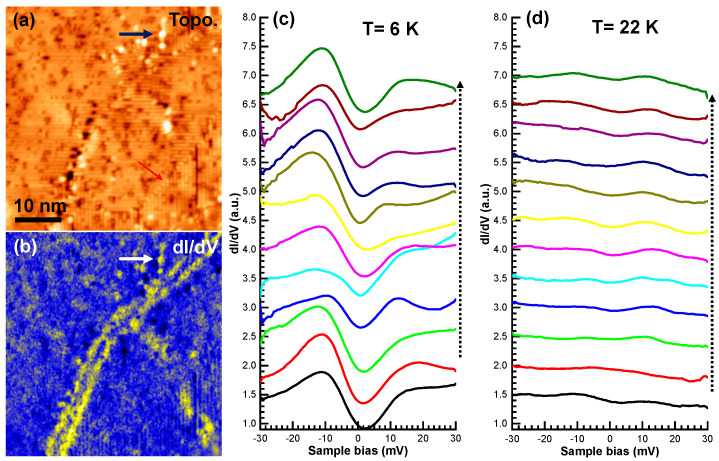
(**a**,**b**) STM image and *dI/dV* conductance map on a Pr-Ca122 region (*Vs* = 300 mV, *It* = 200 pA, *V_ac_* = 1 mV, *f* = 973 Hz, *T* = 6 K). (**c**,**d**) *dI/dV* line spectra along the dotted arrow on the same area at 6 K and 22 K, respectively (*Vs* = −10 mV, *It* = 200 pA, *V_ac_* = 0.5 mV, *f* = 973 Hz). The same vertical offset is applied to the spectra in both (**c**,**d**). The red arrow points to a reconstruction of the domain boundary.

**Figure 5 nanomaterials-11-01019-f005:**
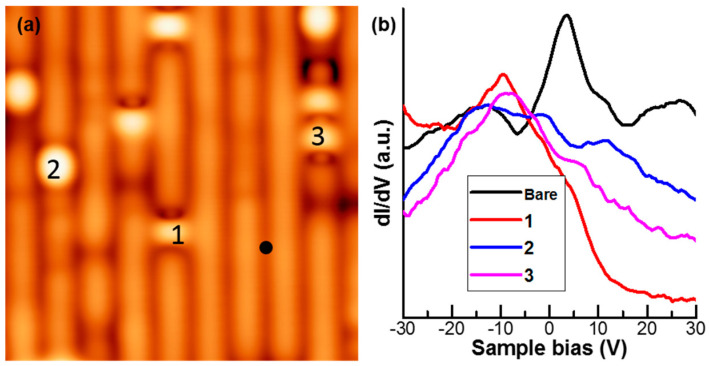
(**a**) STM atomic resolution image in a pristine area of Pr-Ca122 surface with (*V_S_* = −2 mV, *It* = 200 pA, *T* = 4.2 K). (**b**) The *dI/dV* spectra taken at three different pristine defects are very different from one taken at the bare area, and they all show no superconducting gap.

## Data Availability

The data presented in this study are available on request from the corresponding author.

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
