# Peer review of "Nanoscale Superconducting States in the Fe-Based Filamentary Superconductor of Pr-Doped CaFe2As2"

_nanomaterials, 2021, doi:10.3390/nano11041019_

Round 1
Reviewer 1 Report
The article presents a convincing and interesting experimental demostration that a Fe-based filamentary superconductor presents nanoscale-localized robust superconducting states. It also presents compelling evidence supporting that the effect has an intrinsic bulk origin (rather than being induced by, e.g., surface roughness). For that, the main experiments employed are cryogenic STM/STS measurements and local voltage-electrical current characteristics.
The paper is, in my opinion, a relevant contribution to an also relevant subject matter. The supporting experimental work seems to be well executed and convincing. The writting and presentation of the results are also of good quality (the authors use a well-executed and welcome straight-to-the-point style).
As such, I expect the paper to deservely attract a reasonable quantity of citations.
I therefore recommend publication of the paper, without further referee review rounds. I only have a minor suggestion, for voluntary consideration, to the authors: I think it would be better to precisely define what is undertood by the term "filamentary superconductor" in the introductory paragraph (for better convenience of a broader range of readers).
Author Response
Responses:
We thank reviewer for acknowledging the quality of the manuscript.
About the “filamentary superconductor” comment: The majority research in this area focuses on bulk superconducting samples, and only limited studies have been conducted on filamentary superconducting behavior. It has been argued that the filamentary superconductors arise due to the spontaneous electronic inhomogeneity at the nanoscale level of the sample and the STM/S might be the most adequate tool to reveal the insight to filamentary behavior. Such electronic inhomogeneities may be caused by nanoscale dopants and disorder that manifest in non-zero resistance and a very small Meissner effect (field-cooled in magnetic susceptibility), although there is shielding same as bulk superconductors.
Changes:
The above paragraph as well as two references have been added to the introductory paragraph, Line 50-58, reference 5, 6.
Reviewer 2 Report
This paper reports the low-temperature scanning tunneling microscope and spectroscopy for 9% Pr-doped CaFe2As2 to investigate its nanoscale superconducting states. It is successfully shown that local superconducting states emerge near Pr dopants. The results are interesting and reliable, so that the referee believes this work merits publication in this journal after minor revision.
1. I cannot fully understand what new insight is obtained in this study. Refs [3,4] already clarified that the individual Pr dopants responsible for superconductivity. I think that low Pr content (9%) in this study allows to avoid the segregated dopant clusters. However, it seems to be indicated in Refs [3,4].
2. How did the authors decide that the bulk Tc as 22 K?
3. Significant digits of lattice parameters (a = 0.395 nm and c = 1.3 nm) are too small.
4. Did the magnetization measurements performed under constant magnetic field? It is better to note the magnetic field strength.
Author Response
This paper reports the low-temperature scanning tunneling microscope and spectroscopy for 9% Pr-doped CaFe2As2 to investigate its nanoscale superconducting states. It is successfully shown that local superconducting states emerge near Pr dopants. The results are interesting and reliable, so that the referee believes this work merits publication in this journal after minor revision.
- I cannot fully understand what new insight is obtained in this study. Refs [3,4] already clarified that the individual Pr dopants responsible for superconductivity. I think that low Pr content (9%) in this study allows to avoid the segregated dopant clusters. However, it seems to be indicated in Refs [3,4].
Response:
Ref 3 studied the electronic inhomogeneous of the sample. But maybe because of high concentration of Pr content and lack of knowledge of identifying Pr dopant position, they suggested “Our analysis shows that the inhomogeneous and strongly localized high-Tc superconducting state emerges from cloverlike defects, and is a consequence of a Pr distribution”.
Ref 4 provided a very detailed and convinced method to identify Pr dopants on surface. But they did not study local superconductivity behavior. So they just “We therefore suggest that the low volume fraction high-Tc superconducting phase is unlikely to originate from Pr inhomogeneity.”
In this manuscript, we identify all Pr dopant position using method from Ref 4, and clearly plot out local SC map, local density of state map and corresponding atomic resolution images, then make connections among them. The comparison of the spectra taken at low temperature (4.2 K) and an elevated temperature 22 K which is although below but near the bulk transition temperature from the exact same area confirms the local superconducting behavior. We, therefore, suggest that the competition between the local strain at Pr dopants and the local magnetic moment induced by defects is responsible for local superconducting states observed in this Fe based filamentary superconductor.
- How did the authors decide that the bulk Tc as 22 K?
Response:
We apologize for the confusion and thank reviewer for pointing this out. 22 K is the temperature for the set of STS data shown in the manuscript. The bulk Tc is around 30 K. We compared two temperature data (4.2 K and 22 K), one is way below Tc, the other is close to Tc.
Changes:
We have rephrased it in the manuscript: line 29, and line 210.
- Significant digits of lattice parameters (a = 0.395 nm and c = 1.3 nm) are too small.
Response:
Those are lattice constants from literature and crystal database, not from our STM experiment.
- Did the magnetization measurements performed under constant magnetic field? It is better to note the magnetic field strength.
Response:
Yes, the magnetization measurement was performed under a constant magnetic field of 20 Oe.
Change:
The field number is added to Fig. 1 figure caption.